# Elevated brain-derived cell-free DNA among patients with first psychotic episode – a proof-of-concept study

Asael Lubotzky[1,2†], Ilana Pelov[3†], Ronen Teplitz[3], Daniel Neiman[1], Adama Smadja[4], Hai Zemmour[1], Sheina Piyanzin[1], Bracha-Lea Ochana[1], Kirsty L Spalding[5], Benjamin Glaser[6], Ruth Shemer[1]*, Yuval Dor[1]*, Yoav Kohn[3,4]*

[1]Department of Developmental Biology and Cancer Research, Institute for Medical Research Israel-Canada, the Hebrew University-Hadassah Medical School, Jerusalem, Israel; [2]Neuropediatric Unit, Shaare Zedek Medical Center, Jerusalem, Israel; [3]Jerusalem Mental Health Center, Eitanim Psychiatric Hospital, Jerusalem, Israel; [4]Hebrew University-Hadassah School of Medicine, Jerusalem, Israel; [5]Karolinska Institute, Department of Cell and Molecular Biology Stockholm, Stockholm, Sweden; [6]Endocrinology and Metabolism Service, Hadassah Medical Organization and Faculty of Medicine, Hebrew University of Jerusalem, Jerusalem, Israel

*For correspondence:
shemer.ru@mail.huji.ac.il (RS);
yuvald@ekmd.huji.ac.il (YD);
yoavk@ekmd.huji.ac.il (YK)

†These authors contributed equally to this work

**Abstract** Schizophrenia is a common, severe, and debilitating psychiatric disorder. Despite extensive research there is as yet no biological marker that can aid in its diagnosis and course prediction. This precludes early detection and intervention. Imaging studies suggest brain volume loss around the onset and over the first few years of schizophrenia, and apoptosis has been proposed as the underlying mechanism. Cell-free DNA (cfDNA) fragments are released into the bloodstream following cell death. Tissue-specific methylation patterns allow the identification of the tissue origins of cfDNA. We developed a cocktail of brain-specific DNA methylation markers, and used it to assess the presence of brain-derived cfDNA in the plasma of patients with a first psychotic episode. We detected significantly elevated neuron- (p=0.0013), astrocyte- (p=0.0016), oligodendrocyte- (p=0.0129), and whole brain-derived (p=0.0012) cfDNA in the plasma of patients during their first psychotic episode (n=29), compared with healthy controls (n=31). Increased cfDNA levels were not correlated with psychotropic medications use. Area under the curve (AUC) was 0.77, with 65% sensitivity at 90% specificity in patients with a psychotic episode. Potential interpretations of these findings include increased brain cell death, disruption of the blood-brain barrier, or a defect in clearance of material from dying brain cells. Brain-specific cfDNA methylation markers can potentially assist early detection and monitoring of schizophrenia and thus allow early intervention and adequate therapy.

## Editor's evaluation

In this study, the authors have analysed the contribution of brain-derived cell-free DNA in the blood of patients with psychosis based on the established methylation-based tissue deconvolution methodology. They have demonstrated a higher level of brain-derived cell-free DNA in patients who experienced psychotic symptoms compared to healthy controls. The finding would serve as a proof of concept for brain-derived cell-free DNA as biomarkers of psychosis.

## Introduction

### Early diagnosis in schizophrenia

Schizophrenia is a complex and chronic psychiatric disorder affecting around 1% of the global population. It is associated with increased disability, mortality, and high private and social economic burden (*Laursen et al., 2014*; *Chong et al., 2016*; *Insel, 2010*).

Psychosis is the first presentation of schizophrenia that usually leads to the diagnosis. Although psychosis typically onsets during adolescence and young adulthood, there is accumulating data that demonstrates underlying biological changes, beginning years prior to the psychotic symptoms (*Kahn and Sommer, 2015*; *Lewandowski et al., 2011*). Thus, identifying biomarkers that will allow early diagnosis and therapeutic interventions is of the highest importance.

The etiology of schizophrenia is multifactorial, with a complex interaction of polygenic risk and environmental factors (*Insel, 2010*; *Owen et al., 2016*). Although over 100 genetic variants were identified contributing to the risk of schizophrenia, their broad distribution in the general population precludes clinical use (*Stefansson et al., 2009*).

Other potential biomarkers were studied, including neurotransmitters and their metabolites (*Davison et al., 2018*), associated endophenotypes such as smooth pursuit eye movement (SPEM) (*Morita et al., 2017*) and sensory gating defects (P150) (*Suh et al., 2021*). Results of those studies are somewhat inconsistent and are not specific enough to allow their use as a diagnostic tool. Brain imaging has demonstrated notable structural changes mostly in late and more chronic stages of the disease, too late for useful intervention (*McCutcheon et al., 2020*).

### Cell-free DNA as a biomarker for diseases

Circulating cell-free DNA (cfDNA) in the blood plasma was first discovered by Mandel and Metais in 1948 (*Liu, 2018*). Over the years it was established that these typically ~150 bp fragments of double-stranded DNA are released into the bloodstream during the process of cell death (*Lo et al., 2021*). cfDNA molecules are rapidly cleared from the circulation with a half-life of ~15 min, thus reflecting events taking place at the time of blood draw (*Lo et al., 1999*; *Sherwood and Weimer, 2018*). Although the mechanism underlying the release and rapid clearance of the cfDNA is still unclear, many researchers have been trying to develop methods that will enable the use of these short fragments for the diagnosis and follow-up of various medical conditions. For example, cfDNA-based methods of prenatal diagnosis have been developed (*Lo et al., 1997*; *Wong and Lo, 2003*), and ongoing work is advancing approaches for cancer liquid biopsy (*Volckmar et al., 2018*; *Schwarzenbach et al., 2011*; *Corcoran and Chabner, 2018*) and for detection of rejection in organ transplant recipients (*Sherwood and Weimer, 2018*; *Lehmann-Werman et al., 2018*; *Gala-Lopez et al., 2018*).

The possibility of using cfDNA as a biomarker in the medical conditions detailed so far is based on the fundamental genetic difference between the DNA of a host and that of a tumor, fetus, or the graft (*Lehmann-Werman et al., 2018*; *Lehmann-Werman et al., 2016*; *Zemmour et al., 2018*; *Sun et al., 2015*; *Gai et al., 2018*). In comparison it is more complicated to assess changes in the levels of cfDNA in plasma of healthy individuals or individuals without mutated DNA in their blood. It is known that plasma levels of cfDNA vary over time depending on age, increased physical activity, and the existence of various medical conditions such as infectious diseases (*Sherwood and Weimer, 2018*). Therefore, it has not been possible to use the amount of cfDNA in the blood as a marker for a specific pathology. Moreover, although the cfDNA that circulates in the blood originates from different tissues, it has the same genome and therefore cannot be associated with a specific source tissue using DNA sequencing (*Lehmann-Werman et al., 2016*; *Moss et al., 2018*; *Dor and Cedar, 2018*).

We and others have recently described an approach to identify the tissue origins of cfDNA, based on tissue-specific methylation patterns. Such cell-type-specific markers allow the inference of cell death in multiple settings, for example, cardiomyocyte cell death following myocardial infarction or hepatocyte death in patients with liver metastasis (*Lehmann-Werman et al., 2018*; *Lehmann-Werman et al., 2016*; *Zemmour et al., 2018*; *Sun et al., 2015*; *Gai et al., 2018*; *Lubotzky et al., 2022*).

### Brain cell death in psychosis and schizophrenia

Neuro-anatomical changes are known in schizophrenia. Numerous longitudinal and cross-sectional imaging studies recruited individuals at high and ultra-high risk for developing a psychotic illness and demonstrated dynamic brain changes emerging around onset and over the years of schizophrenia.

The studies generally demonstrate gray matter reduction in the frontal and temporo-limbic regions (*Takahashi and Suzuki, 2018*; *Dietsche et al., 2017*; *Pantelis et al., 2007*). Structural brain abnormalities in psychosis occur prior to full blown symptoms and progressively worsen as psychosis develops. Although the underlying causes and the exact timing of the morphologic changes in the brain remain obscure, apoptosis has been proposed as a potential mechanism that could contribute to this progressive pathology. This idea is also supported by post-mortem evidence of regional reduction of neuronal and glia density. Apoptosis in the CNS occurs rapidly, and apoptotic cells are typically cleared within 24 hr. The process occurs without inflammation and does not involve gliosis (*Jarskog et al., 2005*; *Jarskog, 2006*). Ershove et al. demonstrated dysregulation of apoptosis in schizophrenia through considerably increased oxidated levels of cfDNA (*Ershova et al., 2017*). Thus, we hypothesized that brain-derived cfDNA levels increase in patients with acute psychosis and schizophrenia, reflecting ongoing cell death, especially in early and active phases of diseases. We further hypothesized that brain cfDNA levels would be increased in patients, unrelated to treatment with antipsychotic medications.

In this proof-of-concept study we examined plasma samples from patients with a first psychotic episode, in order to assess the potential use of brain cfDNA as a method for early detection and diagnosis of schizophrenia in general and acute psychosis events.

## Results

### Identification of brain methylation markers

Tissue-specific methylation markers were selected after a comparison of extensive genome-wide DNA methylation datasets generated using Illumina Infinium HumanMethylation450k BeadChip array. Datasets used included publicly available methylation profiles from The Cancer Genome Atlas and Gene Expression Omnibus repositories, along with data that we generated locally. Using this comparative analysis we selected 13 genomic loci, which are unmethylated specifically in neurons (four markers), oligodendrocytes (three markers), or astrocytes (three markers) and in all brain-derived cell types (three markers), and methylated in all other examined cell types. Data describing the development and validation of these 10 cell-specific markers (neurons, oligodendrocytes, and astrocytes) were published in a recent paper (*Lubotzky et al., 2022*). To test our in silico predictions regarding the three whole brain markers specificity, we applied bisulfite treatment, multiplex PCR, and next-generation sequencing to determine the methylation status of each marker in a panel of DNA samples obtained from multiple human tissues, as previously described (*Neiman et al., 2020*; *Figure 1—figure supplement 1A*). To determine assay sensitivity, we serially diluted brain DNA into leukocyte DNA, and found that the brain markers allowed the detection of as little as 0.1% brain DNA in a mixture, or just one brain genome in a mixture of a thousand genomes (*Figure 1—figure supplement 1B*).

### Noninvasive detection of brain-derived cfDNA among patients with first psychotic episode

We determined the plasma concentrations of brain-derived cfDNA (including methylation markers of neurons, astrocytes, and oligodendrocytes and whole brain) in 31 healthy controls, and 29 patients with first psychotic episode.

Total levels of cfDNA circulating in plasma were higher in patients with first psychotic episode compared with healthy controls (Mann-Whitney test for controls vs. patients, p-value = 0.017, *Figure 1A*). cfDNA levels from all brain cell combined were higher in patients with first psychotic episode compared with healthy (*Figure 1B and C*). cfDNA levels from each brain cell type separately or from whole brain markers were also significantly elevated in patients (Mann-Whitney test for patients vs. controls, neurons p-value = 0.0013, astrocytes p-value = 0.0016, oligodendrocytes p-value = 0.0129, whole brain markers p-value = 0.0012). Elevated brain-derived cfDNA was seen when measuring either its absolute concentration (*Figure 1D*) or its fraction (*Figure 1—figure supplement 2A-D*). This is consistent with elevated release of brain-derived cfDNA in patients with first psychotic episode, reflecting brain pathology, rather than a non-specific effect.

To determine how well brain-derived cfDNA markers can distinguish patients with first psychotic episode from healthy controls, we generated a receiver operating characteristic (ROC) curve for the combined signal from all brain cell types. The brain cfDNA score was able to identify the plasma

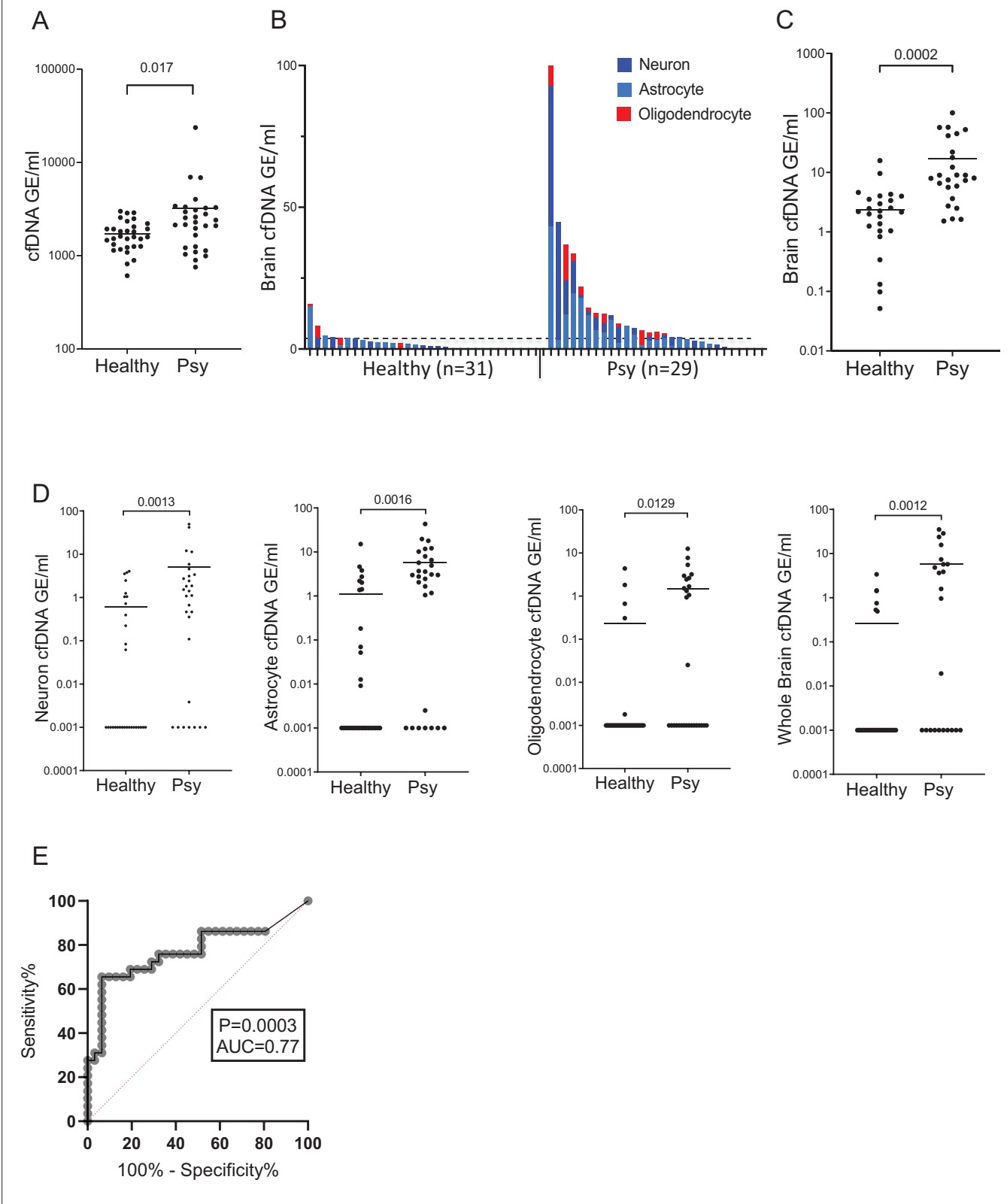

**Figure 1.** Brain-derived cell-free DNA (cfDNA) in patients with a first psychotic episode. (**A**) Plasma concentrations of total cfDNA. Total levels of cfDNA in healthy controls and patients with first psychotic episode. Mann-Whitney test for controls vs. patients, p-value = 0.017. (**B–C**) Plasma concentrations of brain-derived cfDNA. Brain cfDNA levels in healthy controls (n=31) and patients with first psychotic episode (n=29), represented as the cumulative sum of the average signals from neuron, oligodendrocyte, and astrocyte markers. In panel B, graph retaining the value of each component; in panel C, graph

*Figure 1 continued on next page*

*Figure 1 continued*

combining all average signals to one value for statistical analysis. (**D**) Signals from distinct brain cell types. Mann-Whitney test for controls vs. patients, neurons p-value = 0.0013, astrocytes p-value = 0.0016, oligodendrocytes p-value = 0.0129, and whole brain markers p-value = 0.0012. Shown are the average levels in plasma of four neuronal markers, three astrocyte markers, three oligodendrocyte markers, and three general brain markers. cfDNA was treated with bisulfite, PCR amplified, and sequenced, and the molecules that are fully unmethylated were scored. The fraction of brain-specific molecules was multiplied by the total cfDNA concentration. Each dot represents one plasma sample. (**E**) Receiver operating characteristic (ROC) curve for distinguishing patients with first psychotic episode from healthy controls. A combined score of all brain markers, area under the curve (AUC) 0.77; 95% CI=0.64–0.9; p-value = 0.0003.

The online version of this article includes the following figure supplement(s) for figure 1:

**Figure supplement 1.** Specificity and sensitivity of brain methylation markers.

**Figure supplement 2.** Fractions and receiver operating characteristic (ROC) curves for markers of individual brain cell types.

of patients with a first psychotic episode with an area under the curve (AUC) of 0.77 (p-value = 0.0003). At 90% specificity, the sensitivity for identifying patients with first psychotic episode was 65% (*Figure 1E*). The markers of each brain cell type were able to identify the plasma of patients with first psychotic episode with an AUC of 0.65–0.73 (*Figure 1—figure supplement 2E-H*).

There was no significant correlation between brain markers and clinical parameters including the duration of the current episode, duration of hospitalization, or the number of psychotic episodes (see *Source data 1*).

## Relationship to psychiatric drugs

Since previous studies found a correlation between the use of antipsychotic drugs and anatomic changes in the brain (*Huhtaniska et al., 2017*), we examined whether the increase of brain-derived cfDNA levels in patients is related to their psychiatric drug intake. To this effect, we considered (1) the amount of antipsychotic drugs taken on the day of the patient's blood test for cfDNA and (2) the total amount of antipsychotic drugs taken throughout the hospitalization until the blood test. These data were taken from medical records. In order to generate a single score for all patients, independently of the type of drugs they took, we converted the various antipsychotic medication dosages into an equivalent dosage of olanzapine, following the expert-consensus-based method used in *Gardner et al., 2010*. We then ran various statistical tests (Pearson and Spearman correlation coefficients, and t-test and Mann-Whitney for categorical comparisons), to examine potential correlations with the total amount of cfDNA, and the level of cfDNA originating from astrocytes, neurons, oligodendrocytes, and from the whole brain. We considered 13 individual biomarkers for brain cell types, as well as their average for each cell type, for a total of 18 variables.

After correcting for multiple testing, we did not find any statistically significant correlation between drug intake and cfDNA. Also, all statistical tests apart from three were non-significant on a nominal level (p>0.05). We found nominally statistically significant correlations between the levels of brain-derived cfDNA and drug intake for only one type of astrocyte marker (PRDM2) and one type of neuron marker (ITF) (Spearman correlation coefficient for astrocyte marker PRDM2: p-value = 0.024/drugs on blood test day, and p-value = 0.025/overall amount drugs; for neuron ITF: p-value = 0.016/overall amount of drugs).

## Discussion

We have demonstrated increased levels of brain-derived cfDNA in adult patients with a first psychotic episode. This is a proof-of-concept study, demonstrating the feasibility of using cfDNA as a biomarker in psychiatric patients, and the results call for replication in a larger number of well-characterized patient samples, requiring better control for diagnosis and drug treatment. Further research is needed to examine the association of increased brain cfDNA levels with specific diagnoses, mainly schizophrenia, and with other factors such as genetic predisposition and course of disease.

We note that our study detected elevated brain cfDNA but did not reveal the cause of release of these molecules to circulation. We hypothesize that our findings reflect brain cell apoptosis, which is consistent with the literature indicating brain damage in such patients. However, elevated brain cfDNA could also result from a disruption of the blood-brain barrier, from a defect in the local removal of debris from dying brain cells, or a combination of these biological phenomena. Regardless,

elevated brain cfDNA provides an intriguing, measurable link between a physical process in the brain and a psychotic episode. Another alternative driver of cfDNA, which we could not robustly rule out, was psychiatric drug treatment or illicit drug abuse both of which could theoretically induce brain cell death, although we partially controlled for that. Future exploration should aim at distinguishing between these possibilities. We also note that samples were obtained after the initial stabilization and partial remission of the psychotic symptoms, suggesting the possibility that a more dramatic elevation of brain cfDNA takes place during the acute phase of the psychotic episode. Finally, we observed an elevation in the total concentration of cfDNA among psychotic patients. Since brain cfDNA comprises a small fraction of cfDNA even among patients (<1% of total cfDNA), this elevation must originate in other tissues. Elevated total cfDNA can be related to drug toxicity, stress mechanisms, or more likely derive from immune or inflammatory cells, perhaps related to the proposed link between schizophrenia and the immune system (*Zengeler and Lukens, 2021*; *Khandaker et al., 2015*). In a recent paper (*Fox-Fisher et al., 2021*) we characterized cfDNA released from specific immune cell types. We suspect that the total cfDNA elevation originates from these cells and hope to explore that in future research.

The development and validation of brain-specific cfDNA methylation markers can shed light on the biology of schizophrenia and other brain diseases, and potentially assist early detection and monitoring of treatment response. The current literature suggests that underlying biological changes of the disease begin years prior to the psychotic symptoms. Elevated cfDNA can reflect ongoing changes in the brain of patients with schizophrenia in early stages that are too subtle to be seen by imaging techniques. On the other hand, taking into account the short half-life of cfDNA in the plasma and time of blood sampling (a few weeks after admission in most cases), our results suggest that pathological brain processes occur for a longer period than the acute presentation. This is also supported by the literature as brain changes emerging around onset of the schizophrenia and accumulate over the years of the disease. It would be very interesting to sample patients in the earlier stage of psychosis, but indeed will be difficult due to ethical (consent) issues. Our results may also aid in monitoring treatment response. Increased brain-derived cfDNA can be a marker of psychotic exacerbation in chronic schizophrenia. This study was not designed to prove the above-mentioned hypotheses, and this could be a goal for further research.

## Materials and methods
### Clinical samples

We recruited men and women over 18 years of age, who have developed psychotic symptoms for the first time in their life within the last year and were admitted to the acute psychiatric units of the Jerusalem Mental Health Center in Israel. Diagnosis of a psychotic disorder was given by treating psychiatrists and verified by a senior psychiatrist from the research team (IP). Blood samples were obtained after the initial stabilization and partial remission of the psychotic symptoms which enabled the patients to give written informed consent for participation in the study. The time achieving stability was different for each individual patient, but the majority stabilized within 4 weeks (79%). The patients were asked to complete a short questionnaire regarding demographic details, current physical condition, the onset of present symptoms, and drug use. The recruitment was done following the approval of the study's protocol by the Jerusalem Mental Health Center Institutional Review Board. Patients with acute medical conditions in the week prior to the blood sampling were excluded from the study.

A total of 31 patients were recruited between the years 2017 and 2019. DNA extraction from two samples failed, and the remaining 29 samples were analyzed. The study included 55% females (n=16) and 45% males (n=13), with an age distribution of 18–48 years (average 24). 72.5% of the patients (n=21) were hospitalized with a first psychotic event in their life, and 27.5% of the patients (n=8) had a second or third psychotic event, up to a year after the beginning of the first one, which clinically can be considered as the first psychotic event in life that has not reached complete and stable remission. 44.8% of the patients (n=13) did not have any background of drug abuse, 31% (n=9) indicated the use of marijuana in the weeks preceding the onset of the first psychotic episode. 24.2% patients (n=7) described using several different types of drugs (hallucinogens, MDMA, cocaine, etc.) in the past. Clinical data, including medications used, were obtained from medical records.

Thirty-one adult volunteers participated in the study as unpaid healthy controls. They were age and sex matched to hospitalized patients. Controls were screened for any medical acute and chronic issues that can potentially increase levels of cfDNA in plasma. They also were screened for lifestyle factors which can influence the cfDNA levels such as exercising before the blood test.

This study was conducted according to protocols approved by the Jerusalem Mental Health Center Institutional Review Board, in accordance with the Declaration of Helsinki (4-17).

Patient demographics, clinical data, and cfDNA data are detailed in *Source data 1*.

## cfDNA analysis

Blood samples were collected in cfDNA BCT (Streck) tubes and centrifuged at 1500 *g* for 10 min at 25°C within 1 week of collection. Plasma was removed and re-centrifuged at 3000 *g* for 10 min at 25°C to remove any remaining cells. Plasma was then stored at −80°C until assay. Cell-free DNA was extracted using the QIAsymphony SP instrument and its dedicated QIAsymphony Circulating DNA Kit (Qiagen) according to the manufacturer's instructions. DNA concentration was measured using the Qubit dsDNA HS Assay Kit (Thermo Fisher Scientific). cfDNA was treated with bisulfite using EZ DNA Methylation-Gold (Zymo Research), and PCR amplified with primers specific for bisulfite-treated DNA but independent of methylation status at the monitored CpG sites, as described (*Neiman et al., 2020*). Treatment with bisulfite led to degradation of 60–90% of the DNA (average, 75% degradation), consistent with previous reports (*Grunau et al., 2001*). Note that while DNA degradation does reduce assay sensitivity (since fewer DNA molecules are available for PCR amplification), it does not significantly harm assay specificity since methylated and unmethylated molecules are equally affected. Primers were bar-coded using TruSeq Index Adapters (Illumina), allowing the mixing of samples from different individuals when sequencing PCR products using NextSeq sequencers (Illumina). Sequenced reads were separated by barcode, aligned to the target sequence, and analyzed using custom scripts written and implemented in R. Reads were quality filtered based on Illumina quality scores, and identified by having at least 80% similarity to target sequences and containing all the expected CpGs in the sequence. CpGs were considered methylated if CG was read and were considered unmethylated if TG was read. The fraction of unmethylated molecules in a sample was multiplied by the total concentration of cfDNA in the sample, to assess the number of brain genome equivalents per ml of plasma. The concentration of cfDNA was measured prior to bisulfite conversion, rendering the assay robust to potential inter-sample fluctuations in the extent of bisulfite-induced DNA degradation.

The computational pipeline used to interpret sequence reads as well as a representative set of data were uploaded to GitHub (https://github.com/Joshmoss11/btseq; copy archived at swh:1:rev:ef-c75ddd347c20392cf0a034706a7b5b6090be75, *Moss, 2021*).

## Data availability

All relevant data including information on the markers used (coordinates and primer sequences), detailed information on patients and donors, and the raw data on values of each methylation marker in each sample are provided in *Source data 1*. These data were used to generate the graphs shown in the paper.

Detailed PCR conditions are detailed in the Materials and methods section and were published in a recent paper (*Neiman et al., 2020*). Code is uploaded to GitHub as described in the paper.

## Acknowledgements

The authors thank all patients and controls for their participation in the study. Funding supported by a Joint Award of the National Institute of Psychobiology in Israel (NIPI) and the Israeli Society of Biological Psychiatry (to IP and AL). Supported by a grant from Alzheimer's Drug Discovery Foundation (ADDF) (to AL and YD). Supported by grants from Grail, The Ernest and Bonnie Beutler Research Program of Excellence in Genomic Medicine, The Israel Science Foundation, the Grant for Multiple Sclerosis Innovation (GMSI) from Merck, the Waldholtz/Pakula family, the Robert M and Marilyn Sternberg Family Charitable Foundation (to YD). YD holds the Walter and Greta Stiel Chair and Research grant in Heart studies.

# Additional information

### Competing interests
Daniel Neiman, Hai Zemmour, Benjamin Glaser, Ruth Shemer, Yuval Dor: has filed patents on cfDNA analysis technology (year 2019, application number 62/828,587). The other authors declare that no competing interests exist.

### Funding

| Funder | Grant reference number | Author |
|---|---|---|
| Award of the National Institute of Psychobiology in Israel and the Israeli Society of Biological Psychiatry | | Asael Lubotzky Ilana Pelov |
| Israel Science Foundation | | Yuval Dor |
| Grail | | Yuval Dor |
| Alzheimer's Drug Discovery Foundation | | Asael Lubotzky Yuval Dor |
| Merck | Grant for Multiple Sclerosis Innovation [GMSI] | Yuval Dor |

The funders had no role in study design, data collection and interpretation, or the decision to submit the work for publication.

### Author contributions
Asael Lubotzky, Conceptualization, Data curation, Formal analysis, Methodology, Project administration, Writing – original draft; Ilana Pelov, Investigation, Project administration, Writing – review and editing; Ronen Teplitz, Sheina Piyanzin, Bracha-Lea Ochana, Kirsty L Spalding, Investigation; Daniel Neiman, Investigation, Methodology; Adama Smadja, Formal analysis; Hai Zemmour, Methodology; Benjamin Glaser, Supervision, Writing – review and editing; Ruth Shemer, Conceptualization, Investigation, Methodology, Supervision, Writing – review and editing; Yuval Dor, Conceptualization, Investigation, Methodology, Resources, Writing – review and editing; Yoav Kohn, Conceptualization, Supervision, Writing – review and editing

### Author ORCIDs
Asael Lubotzky (ID) http://orcid.org/0000-0002-0460-0084
Benjamin Glaser (ID) http://orcid.org/0000-0003-4711-5000
Yuval Dor (ID) http://orcid.org/0000-0003-2456-2289
Yoav Kohn (ID) http://orcid.org/0000-0003-0476-9610

### Ethics
This study was conducted according to protocols approved by the Jerusalem Mental Health Center Institutional Review Board, in accordance with the Declaration of Helsinki (4-17). We recruited men and women over 18 years of age, who have developed psychotic symptoms for the first time in their life within the last year and were admitted to the acute psychiatric units of the Jerusalem Mental Health Center in Israel. Blood samples were obtained after the initial stabilization and partial remission of the psychotic symptoms which enabled the patients to give written informed consent for participation in the study. The time achieving stability was different for each individual patient, but the majority stabilized within 4 weeks (79%). The patients were asked to complete a short questionnaire regarding demographic details, current physical condition, the onset of present symptoms and drug use. The recruitment was done following the approval of the study's protocol by the Jerusalem Mental Health Center Institutional Review Board. Patients with acute medical conditions in the week prior to the blood sampling were excluded from the study. 31 adult volunteers participated in the study as unpaid healthy controls.

### Decision letter and Author response
Decision letter https://doi.org/10.7554/eLife.76391.sa1

Author response https://doi.org/10.7554/eLife.76391.sa2

## Additional files

### Supplementary files
• Source data 1.
• Transparent reporting form

### Data availability
All relevant data including information on the markers used (coordinates and primer sequences), detailed information on patients and donors and the raw data on values of each methylation marker in each sample are provided in Source data 1. These data were used to generate the graphs shown in the paper. Detailed PCR conditions are detailed in the Materials and methods section and were published in a recent paper. Code is uploaded to GitHub as described in the paper (https://github.com/Josh-moss11/btseq, copy archived at swh:1:rev:efc75ddd347c20392cf0a034706a7b5b6090be75).

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
