## [Editor Report]

In this study, the authors have analysed the contribution of brain-derived cell-free DNA in the blood of patients with psychosis based on the established methylation-based tissue deconvolution methodology. They have demonstrated a higher level of brain-derived cell-free DNA in patients who experienced psychotic symptoms compared to healthy controls. The finding would serve as a proof of concept for brain-derived cell-free DNA as biomarkers of psychosis.

---

## [Decision Letter]

**Decision letter after peer review:**

Thank you for submitting your article "Elevated brain-derived cell-free DNA among patients with first psychotic episode -a proof-of-concept study" for consideration by *eLife*. Your article has been reviewed by 2 peer reviewers, including Jacky Lam as Reviewing Editor and Reviewer #1, and the evaluation has been overseen by YM Dennis Lo as the Senior Editor.

Essential revisions:

1. Control subjects

Currently age- and sex-matched healthy subjects were used as controls. However, the clinical details of these control subjects were lacking. How the volunteers had been recruited i.e from other clinics and whether they possess any other medical conditions may have implications on result interpretation. Specifically, it is uncertain whether the authors are claiming this test is helpful for diagnosing schizophrenia or risk of developing acute psychosis. If the latter then one might suggest a control group with chronic stable schizophrenia or with the risk factors of schizophrenia (e.g. chronic alcohol use or illicit drug use) might be more appropriate.

2. Clarification on the timing of blood sampling

In the study, blood samples were obtained 'after initial stabilisation and partial remission of psychotic symptoms', with the consideration of the informed consent issue. The description of the timing of blood sampling is vague. It is unclear how soon after admission the patients were sampled. Was this on admission i.e during the 'acute psychotic episode' or later once the patient had been stabilised? One presumes the latter (for consent purposes) which makes the findings of cfDNA presence surprising given the 15 minute half-life of cfDNA quoted.

Also, the authors are suggested to provide the biological rationale for 'acute' sampling if MRI changes in schizophrenia imply chronic apoptosis.

3. Heterogeneity of patients with schizophrenia

As highlighted by the authors, schizophrenia is multifactorial and complex. Psychotic disorders could be classified into different subgroups according to the disease presentation and/or aetiology, e.g. alcohol-induced schizophrenia. The authors are suggested to include a more detailed description of the clinical characteristics of the schizophrenic patients and also the grouping. The alcohol use history should also be included, in addition to the drug abuse/ marijuana use history.

4. The authors are suggested to propose how this test can lead to early detection and how it might be used for monitoring treatment response.

5. The authors are suggested to provide evidence for the origin of the increase in total cfDNA in patients with an acute psychotic episode – could this be related to treatment e.g systemic tissue damage e.g due to injury/restraint/multiple venesections?

6. The use of anti-psychotic drugs is covered in its own Results section however the paper would benefit from a table identifying these medications. A distinction should also be made between anti-psychotic medications and other medications used in the acute psychotic episode eg benzodiazepines.

---

## [Author Response]

Essential revisions:1. Control subjectsCurrently age- and sex-matched healthy subjects were used as controls. However, the clinical details of these control subjects were lacking. How the volunteers had been recruited i.e from other clinics and whether they possess any other medical conditions may have implications on result interpretation. Specifically, it is uncertain whether the authors are claiming this test is helpful for diagnosing schizophrenia or risk of developing acute psychosis. If the latter then one might suggest a control group with chronic stable schizophrenia or with the risk factors of schizophrenia (e.g. chronic alcohol use or illicit drug use) might be more appropriate.

Thank you. The clinical details of the control healthy subjects are detailed in Source data 1. The samples were recruited from healthy volunteers working or visiting the hospital (including students). The volunteers were screened for any medical acute and chronic issues that can potentially increase levels of cfDNA in plasma. They also were screened for lifestyle factors which can influence the cfDNA levels like exercising before the blood test. This is detailed now in the Materials and methods Section of the manuscript (Page 13, first paragraph).

We suggest that our test can, in the future, assist early detection of schizophrenia because current literature suggests that underlying biological changes of the disease will begin years prior to the psychotic symptoms. We suggest that this test can assist in early diction as reflection of ongoing changes in the brain that are too subtle to be seen by imaging technics and maybe aid in monitoring of treatment response. The hypothesis that increased brain-derived cfDNA is a marker of psychotic exacerbation in chronic schizophrenia is interesting. However, this study was not designed to prove it, and this could be a goal for further studies This is detailed now in the Discussion section of the manuscript.

2. Clarification on the timing of blood samplingIn the study, blood samples were obtained 'after initial stabilisation and partial remission of psychotic symptoms', with the consideration of the informed consent issue. The description of the timing of blood sampling is vague. It is unclear how soon after admission the patients were sampled. Was this on admission i.e during the 'acute psychotic episode' or later once the patient had been stabilised? One presumes the latter (for consent purposes) which makes the findings of cfDNA presence surprising given the 15 minute half-life of cfDNA quoted.

Blood samples were obtained after the initial stabilization and partial remission of the psychotic symptoms which enabled the patients to give written informed consent for participation in the study. The time achieving stability is different for each individual patient, but the majority stabilized within 4 weeks (79%). This is explained now in the Materials and methods Section of the manuscript.

Taking in account the short half-life of cfDNA in the plasma and time of blood sampling (a few weeks after admission in most cases), our results suggest that pathological brain processes occur for a longer period than the acute presentation. This is also supported by literature as brain changes emerging around onset of the schizophrenia and accumulate over the years of the disease. It would be very interesting to sample patients in the earlier stage of psychosis, but indeed will be difficult due to ethical (consent) issues. This is detailed now in the Discussion section of the manuscript.

Also, the authors are suggested to provide the biological rationale for 'acute' sampling if MRI changes in schizophrenia imply chronic apoptosis.

Thanks for clarifying. Structural brain abnormalities in psychosis occur prior to full blown symptoms and progressively worsen as psychosis develops. Although the underlying causes and the exact timing of the morphologic changes in the brain remain obscure, apoptosis has been proposed as a potential mechanism that could contribute to this progressive pathology. It is reasonable to think that this process peaks near the appearance of clinical symptoms, however further research should address the timing curve. This is also discussed in our reply above and added to the text as mentioned.

3. Heterogeneity of patients with schizophreniaAs highlighted by the authors, schizophrenia is multifactorial and complex. Psychotic disorders could be classified into different subgroups according to the disease presentation and/or aetiology, e.g. alcohol-induced schizophrenia. The authors are suggested to include a more detailed description of the clinical characteristics of the schizophrenic patients and also the grouping. The alcohol use history should also be included, in addition to the drug abuse/ marijuana use history.

Thank you for the suggestion. We now give detailed clinical information in Source data 1, (in the ‘Clinical data – Patients’ tab) including on drug abuse. Unfortunately, we have no data on alcohol consumption prior to hospitalization, but we can be certain that alcohol was not consumed by patients from time of admission to blood sampling (a few weeks later in most cases). Alcohol is not known to affect the levels of cfDNA (Yuwono NL, Warton K, Ford CE. The influence of biological and lifestyle factors on circulating cell-free DNA in blood plasma. eLlife. 2021). Also, it should be noted that unlike cannabis, alcohol is not a strong risk factor for psychosis in general, and in schizophrenia in particular. Regarding the grouping, due to heterogeneity and a small sample size it’s hard to divide a group of 29 patients to sub-groups. This is a proof-of-concept study which will require further research.

4. The authors are suggested to propose how this test can lead to early detection and how it might be used for monitoring treatment response.

Thank you very much for highlighting this point. Large scale validation of our test can find these markers as a useful tool for identifying wide range of psychiatric diseases, perhaps even when only mild symptoms appear. In addition, the test can serve as a biomarker for treatment response, for example, by monitoring the levels of brain derived cfDNA as a sign for remission or relapse. This is detailed now in the Discussion section of the manuscript.

5. The authors are suggested to provide evidence for the origin of the increase in total cfDNA in patients with an acute psychotic episode – could this be related to treatment e.g systemic tissue damage e.g due to injury/restraint/multiple venesections?

As can be seen in Graph1A, there is an elevation in the total amount of cfDNA patients with first psychotic episode (p=0.017). Elevated brain derived cfDNA was seen when measuring either its absolute concentration or its fraction, reflecting brain pathology rather than a non-specific effect. Yet, the elevation in total cfDNA probably originates from other tissues since brain cfDNA comprises less than 1 percent of the total cfDNA. This elevation can be related to drug toxicity, stress mechanisms or more likely derive from immune cells. In a recent paper (Fox-Fisher, Ilana et al. “Remote immune processes revealed by immune-derived circulating cell-free DNA.” *eLife* 2021) we characterized cfDNA released from specific immune cell types. We suspect that the total cfDNA elevation originates from these cells and hope to explore that in future research. This is detailed now in the Discussion section of the manuscript.

6. The use of anti-psychotic drugs is covered in its own Results section however the paper would benefit from a table identifying these medications. A distinction should also be made between anti-psychotic medications and other medications used in the acute psychotic episode eg benzodiazepines.

Detailed tables of the different medications used by patients are given now in Source data 1.